# The Cortical Silent Period and Its Association with Fatigue in Multiple Sclerosis: The Need for Standardized Data Collection

**DOI:** 10.3390/brainsci14010028

**Published:** 2023-12-26

**Authors:** Sebastian Strauss, Thorsten Herr, Christina Nafz, Nelly Seusing, Matthias Grothe

**Affiliations:** Department of Neurology, University Medicine of Greifswald, 17475 Greifswald, Germany; sebastian.strauss@med.uni-greifswald.de (S.S.); nelly.seusing@med.uni-greifswald.de (N.S.)

**Keywords:** multiple sclerosis, transcranial magnetic stimulation, evoked potentials, electromyography, neurophysiology, motor cortex, fatigue, brain, humans, reproducibility of results

## Abstract

The cortical silent period (CSP), assessed with transcranial magnetic stimulation (TMS), provides insights into motor cortex excitability. Alterations in the CSP have been observed in multiple sclerosis (MS), although a comparison of the sometimes contradictory results is difficult due to methodological differences. The aim of this study is to provide a more profound neurophysiological understanding of fatigue’s pathophysiology and its relationship to the CSP. Twenty-three patients with MS, along with a matched control group, underwent comprehensive CSP measurements at four intensities (125, 150, 175, and 200% resting motor threshold), while their fatigue levels were assessed using the Fatigue Scale for Motor and Cognitive Functions (FSMC) and its motor and cognitive subscore. MS patients exhibited a significantly increased CSP duration compared to controls (*p* = 0.02), but CSP duration was not associated with the total FSMC, or the motor or cognitive subscore. Our data suggest a systematic difference in MS patients compared to healthy controls in the CSP but no association with fatigue when measured with the FSMC. Based on these results, and considering the heterogeneous literature in the field, our study highlights the need for a more standardized approach to neurophysiological data collection and validation. This standardization is crucial for exploring the link between TMS and clinical impairments in diseases like MS.

## 1. Introduction

Multiple sclerosis (MS) is a chronic, autoimmune disease of the central nervous system, characterized by demyelination and axonal damage [1,2]. It affects millions of people worldwide, causing a wide range of physical and cognitive impairments. One of the most prevalent symptoms of MS is fatigue, with up to 90% of the patients being affected [3]. Fatigue in MS is not merely a feeling of tiredness; it is often described as overwhelming exhaustion that significantly impacts daily functioning and the quality of life [3,4]. Despite its prevalence and impact, measuring fatigue objectively has remained a challenge in the field of MS research [5], especially since there is a strong overlap between depression and sleepiness as well as associations with other psychiatric disorders [6,7]. Notably, fatigue in MS often involves both abnormal cognitive and motor fatigability, which can be assessed using the established Fatigue Scale for Motor and Cognitive Functions (FSMC) [8]. The underlying pathophysiological cause of fatigue in MS still remains unclear, with metabolic, muscle-related, and peripheral and central neural components [9,10]. Limited data also suggest that motor fatigue, at least in part, is a cortical phenomenon, demonstrating alterations in motor cortex excitability as assessed by transcranial magnetic stimulation (TMS) [11].

TMS is a noninvasive neurostimulation technique that uses electromagnetic induction to modulate neural activity in specific brain regions [12]. It has gained widespread recognition in the field of neuroscience for its ability to explore and manipulate cortical excitability, providing valuable insights into various neurological and psychiatric conditions [13,14,15]. TMS works by generating brief magnetic fields through a coil placed on the scalp, inducing electric currents in targeted brain areas. This controlled stimulation allows researchers to investigate neural pathways, synaptic connections, and excitability thresholds. In the context of MS, TMS offers a unique opportunity to explore the underlying neural mechanisms of different clinical symptoms like fatigue [16,17,18].

Several neurophysiological measurements have been shown to be associated with fatigue in MS patients so far: the global abnormality of motor-evoked potentials (MEPs) [19], MEP amplitude [20,21], MEP latency [22], the short intracortical inhibition (SICI) [23,24], and the CSP [25,26]. The results for each of these variables are partially contradictory, which at least in part is due to the high methodological heterogeneity of TMS studies, especially in neurological conditions [27]. Out of the different TMS parameters, the CSP is most frequently investigated [22,24,25,26]. The CSP itself is induced over the primary motor cortex and can be achieved with a TMS pulse during the tonic activation of the target muscle, leading to a pause in EMG activity up to 300 milliseconds [12,28]. The CSP reflects both cortical [29] and spinal inhibitions, involving elements like Renshaw cells and IA inhibitory interneurons [30], and corresponds to the temporary suspension of voluntary muscle activity in a specific muscle [28]. The duration of this CSP, in particular, provides insights into motor cortex excitability and the balance between inhibitory and excitatory mechanisms in the central nervous system [28]. Alterations in the CSP have been demonstrated in several neurological diseases, suggesting its high diagnostic potential [14].

In the field of MS, changes in the duration of the CSP have been observed, with contradictory results of unaltered, increased, and decreased duration compared to healthy controls [17,25,31,32]. Additionally, Nantes et al. revealed an association between prolonged CSP and upper limb motor impairment [33]. Alterations in the CSP have also been related to imaging parameters, revealing an association with the volume of white matter lesion load [33] or the cortical magnetization transfer ratio (MTR) in MS [34].

In the neurophysiological investigations between fatigue and the CSP in particular, the results of the various studies are somewhat contradictory: Chalah et al. compared a group of MS patients with and without a clinical impairment based on a subjective fatigue questionnaire and did not find any difference in CSP duration [24]. In contrast, a correlation between the duration of the CSP and the degree of fatigue was demonstrated in a study by Chaves et al. in which a visual analog scale was used to assess task-related fatigue [26]. This task-related increase in CSP duration is in line with the findings from another study by Santarnecchi et al. [25]. This study also revealed that CSP duration, which initially was reduced compared to healthy controls, could be increased after a drug-based therapy with amantadine, leading to normalization in CSP duration [25]. In the heterogeneous literature on the association between neurophysiology and fatigue, it should also be noted that the TMS methods used differ even for the same outcome variable [27]. CSP studies differ to varying degrees, for example, in the stimulus intensity, which reduces the comparability of the results [28]. This methodological problem is well known and has recently been addressed by a number of guidelines [14].

An investigation of the association between CSP duration and the widely utilized Fatigue Scale for Motor and Cognitive Functions (FSMC) has been an unexplored area of research thus far. This issue appears important to investigate, not only because the FSMC is an internationally validated instrument to assess multiple sclerosis-related fatigue and is routinely employed in clinical practice for monitoring disease progression but also due to its unique ability to differentiate between cognitive and physical fatigue [8]. Consequently, a further investigation could provide valuable insights into the intricate pathophysiology of motor fatigue, particularly within the context of multiple sclerosis.

This study aims to achieve two primary objectives. Firstly, it seeks to confirm the disparities in CSP duration between patients diagnosed with multiple sclerosis and a healthy control group. This comparative analysis serves as a fundamental step toward understanding the neurological basis of fatigue in MS. Secondly, the study attempts to demonstrate a clear relationship between the duration of the CSP and FSMC levels in people with multiple sclerosis. By investigating this relationship, we hope to provide insight into the complex interplay between cortical interhemispheric excitability and the subjective experience of fatigue as described by the FSMC. This multifaceted approach not only deepens our understanding of the neural mechanisms involved in MS-related fatigue but also provides an objective perspective on how cognitive and physical aspects of fatigue are represented at the cortical level, which can pave the way for more targeted and effective interventions in the future.

## 2. Materials and Methods

### 2.1. Participants

For this study, 23 RRMS patients (14 female) aged 43.0 ± 10.7 years (mean ± SD) were consecutively recruited from the MS outpatient clinic at the University Medicine of Greifswald. Twenty-one subjects were right-handed, confirmed using the Edinburgh Handedness Inventory. The median Expanded Disability Status Scale (EDSS) score was 1.5 (range: 0–6). The mean BDI value was 9.24 (range: 2–28). Only two patients showed moderate clinically relevant depressive symptoms, as indicated by Beck’s Depression Inventory (score > 19). Patients with other neurological or psychiatric diseases as well as potential interacting comedication (i.e., antispastic or anticonvulsive medication) were excluded.

An age- and gender-matched group of twenty-three healthy participants served as the control group. For details, see Table 1.

All subjects gave informed consent to the experimental procedures, which were approved by the local ethics committee at the University Medicine of Greifswald (BB138/19). The study was performed in accordance with the Declaration of Helsinki and included two sessions: a session with a structural MRI and a second session involving transcranial magnetic stimulation.

### 2.2. Transcranial Magnetic Stimulation

TMS measurements were performed according to the current guidelines [14,15,28,35]. All participants underwent a structural T1-weighted high-resolution MRI imaging and a neuronavigated TMS measurement.

Neuronavigated TMS was performed using frameless stereotaxic neuronavigation with an infrared optical-tracking Polaris camera (Polaris System, Northern Digital, Waterloo, ON, Canada) and BrainSight (BrainSight TMS, Rogue Research Inc., Cambridge, USA). The structural image used was a T1-weighed high-resolution MR scan of the subject’s brain (3T MRI, Siemens Health Care Diagnostics GmbH, Erlangen, Germany). The individual scan and the subjects’ heads were thoroughly coregistered in the Polaris reference frame. After the registration, the left M1 was localized by identifying the ‘hand knob’ as an anatomical landmark for the motor hand area. Starting at the ‘hand knob’, the first dorsal interosseous hotspot was located.

The TMS was delivered using a MagStim Bistim 200 stimulator (MagStim Company Ltd., Whitland, UK) with a monophasic waveform. A figure-eight coil was held by the experimenter with the handle pointing posterior–laterally at ca. 45° to the midline, so the induced current in the brain flowed in an anteromedial direction.

The hotspot for eliciting contralateral MEP in the first dorsal interosseus was defined as the location where the TMS evoked the largest contralateral MEP in the resting muscle. The resting motor threshold was defined as the minimum intensity to elicit 5 out of the 8 MEPs, with a peak-to-peak amplitude of ≥50 μV. To measure the CSP, ten pulses at four different stimulus intensities (120, 150, 175, and 200% RMT) [36] were presented while participants performed a pinch grip at 50% maximal voluntary contraction measured by a dynamometer (AdInstruments Grip force transducer MLT004, AdInstruments, Sydney, Australia), which provided direct feedback during the TMS session [37]. Parameters were chosen to guarantee recommended data consistency [28].

### 2.3. Data Evaluation

All electrophysiological data were stored for offline analysis and analyzed with Signal software v6.04 (Cambridge Electronic Design, Cambridge, UK).

The RMT was recorded as the maximal stimulator output (0–100%). MEP latencies and amplitudes were assessed during measurements and averaged over all stimulus intensities for further analysis. CSP onset was defined as the time point when post-contralateral MEP EMG fell below prestimulus EMG for at least 10 ms. CSP offset was defined as the time point when the EMG activity returned to sustained EMG activity for at least 10 ms [28]. Onset and offset were determined by visual inspection using a horizontal line marking the mean of the rectified EMG before the TMS stimulus ± 2 SD. The prestimulus EMG was measured from −60 ms to −20 ms before the TMS stimulus. CSP duration was subsequently calculated as CSP offset – CSP onset. To further control for cortical excitability, a ratio of individual MEP amplitude and the duration of the corresponding CSP was calculated following Orth and Rothwell [38].

Background EMG was kept constant over different conditions.

### 2.4. Assessment of Fatigue

All patients underwent assessment using the Fatigue Scale for Motor and Cognitive Function (FSMC) [8]. This comprehensive self-report questionnaire comprises 20 items and evaluates both cognitive and physical fatigue across two subscales, with a maximum score of 50 for each subscale and 100 in total. Participants can choose from five response options ranging from “strongly disagree” to “strongly agree”. In terms of total fatigue, scores greater than or equal to 43 were categorized as “mild”, scores greater than or equal to 53 were considered “moderate”, and scores greater than or equal to 63 were considered “severe” fatigue. For the cognitive subscore, scores greater than or equal to 22 were categorized as “mild cognitive fatigue”, scores greater than or equal to 28 were designated as “moderate cognitive fatigue”, and scores greater than or equal to 34 were considered “severe cognitive fatigue”. For the motor subscore, scores greater than or equal to 22 were categorized as “mild motor fatigue”, scores greater than or equal to 27 were considered “moderate motor fatigue”, and scores greater than or equal to 32 were considered “severe motor fatigue” [8].

### 2.5. Assessment of Depressive Symptoms

The severity of depressive symptoms was evaluated using the BDI [39], which is a widely accepted self-report instrument for assessing depressive symptoms and has been validated across various conditions, including multiple sclerosis (MS).

### 2.6. Statistics

Data were normally distributed as determined with visual inspection as well as Kolmogorov–Smirnov tests, so we used parametric statistical procedures. A repeated-measure (rm) ANOVA with “stimulus intensity” (four intensities) considered both a within-group and between-group factor (healthy controls vs. patients) was used to test the first hypothesis. To further investigate the effect of stimulus intensity, separated repeated-measure ANOVAs (factor: stimulus intensity) were calculated for patients and healthy controls followed by pairwise comparisons between different intensities.

To explore the relationship between the CSP and fatigue, several Pearson correlations were determined with the duration of the CSP and motoric, cognitive, and total FSMC values separately. Statistical analysis was performed in SPSS Statistics 22 (IBM, Armonk, NY, USA).

## 3. Results

There were no significant differences in demographic (age and sex), electrophysiological (resting motor threshold, MEP latencies, and MEP amplitudes), and behavioral (maximal voluntary contraction) baseline measures. In our MS cohort, the total FSMC score was 52.3 in mean, which corresponded to mild-to-moderate fatigue. The cognitive FSMC subscore was 27.1, and the motor FSMC subscore was 25.1, again corresponding to mild-to-moderate fatigue. As expected, fatigue measurements and the BDI were highly associated (FSMC total: r = 0.721, FSMC motor subscore: r = 0.664, and FSMC cognition subscore r = 0.727; all *p* ≤ 0.001).

CSP duration increased significantly with increasing stimulus intensity both in the MS patients and the healthy controls (MS patients: 85 ms (SD 49 ms), 115 ms (53 ms), 133 ms (59 ms), and 149 ms (478 ms); healthy controls: 51 ms (28 ms), 99 ms (35 ms), 114 ms (30 ms), and 122 ms (40 ms) at 125%, 150%, 175%, and 200% RMT, respectively; see Figure 1 and Figure 2).

The rm ANOVA revealed longer durations of the CSP in patients than in healthy controls independently from the stimulus intensity (main effect of group: F(1) = 11.45; *p* = 0.02). In both groups, CSP duration increased with the stimulus intensity (main effect of stimulus: F(3) = 109.68, *p* < 0.001). There was no significant interaction between both factors (F(3) = 0.28; not significant).

When considering the groups separately, the rm ANOVA (with the factor considered the stimulus intensity) revealed a significant effect of the stimulus intensity on CSP duration in patients (F(3) = 61.39; *p* < 0.001) and healthy controls (F(3) = 52.98, *p* < 0.001). Post hoc pairwise comparisons revealed significant differences between the different stimulus intensities (each *p* < 0.05).

Correlation analysis showed no significant association between the CSP at each intensity and FSMC values, including motor and cognitive subscores; further details are provided in Table 2. Additionally, no significant correlation was found for the association between the CSP and BDI scores (each *p* > 0.05).

In summary, a general prolonged duration of the CSP in MS patients compared to healthy controls could be demonstrated irrespective of the stimulus intensity. Additionally, CSP duration did not correlate with the subjective fatigue questionnaire scores in our cohort of patients with mild-to-moderate fatigue according to the FSMC.

## 4. Discussion

This study clearly reveals the pathological changes in the neurophysiological parameter of the cortical silent period (CSP) in patients with multiple sclerosis (MS) compared to healthy individuals, a phenomenon that is the same at different stimulus intensities. Surprisingly, in our cohort of MS patients with mild-to-moderate fatigue, no clear association was found between the degree of fatigue and the duration of the CSP. This observation also applied when accounting for the excitability of the corticospinal tract, specifically considering the balance between inhibition and excitability [38]. This unexpected lack of association, which contradicts some other previous studies, underscores the critical need for standardized neurophysiological testing. These findings, as well as further studies, are important not only to decipher the intricate pathophysiological changes in MS but also to understand the complexity of fatigue in this patient group, as well as in other diseases such as chronic fatigue syndrome [40,41]. The fact that, despite clear neurophysiological changes, no direct link was established between the CSP and fatigue indicates the multifaceted nature of MS-related fatigue, which requires further in-depth investigation. Therefore, it is imperative to conduct more rigorous and standardized neurophysiological investigations to fully decipher the underlying mechanisms contributing to fatigue in MS. This will not only improve our understanding of the pathophysiology of MS but will also enable the development of more targeted interventions to treat the various aspects of fatigue in MS patients.

Multiple sclerosis is a multifaceted neurological disorder characterized by a wide range of clinical impairments, among which neuropsychological deficits stand out prominently [42,43] and are sometimes called “hidden symptoms” [5]. These neuropsychological impairments can be subtle and challenging to detect in their early stages, often masquerading as stress- or aging- or seasonal-related processes [44,45,46]. However, these impairments can have a significant impact on an individual’s ability to maintain employment, manage personal relationships, and engage in social activities, and they are even present in the so-called benign courses of MS [47,48,49,50].

Fatigue, as one of the most prevalent “hidden symptoms” of multiple sclerosis, has a high impact on the quality of life, employment status, capacity to work, and sick leave [4]. Recent studies demonstrated not only a strong overlap of fatigue and depression but also a mutually reinforcing relationship between fatigue and depression contributing to a diminished quality of life [6,51]. Therefore, an objective diagnostic approach is crucial to identify fatigue and to enable patient monitoring and response to interventions. So far, the existing assessment tools are subjective and reliant on self-reporting and clinical observation [4,8,52,53]. Incorporating objective measures, such as advanced digital, neuroimaging, and neurophysiological techniques or biomarkers, could revolutionize the diagnostic approaches in the assessment of fatigue in multiple sclerosis and other pathological alterations but have so far only been used on the basis of individual studies [16,18,52,54,55]. An objective diagnostic approach not only enhances the accuracy of identifying neuropsychological impairments but also facilitates the development of new targeted interventions, thereby significantly improving the overall management and quality of life for individuals living with MS.

In the field of neurophysiology, several studies investigated the differences in neurophysiological parameters between MS patients and healthy controls (for an overview, see [17,56]). With respect to the CSP as a measurement of motor cortex excitability and the balance between inhibitory and excitatory mechanisms, in particular, the results are somewhat heterogenous, as studies revealed an increased [25,26], decreased [25], or equal [24,32] CSP duration. In interpreting the results of these studies, it is important to note that different measurement parameters were chosen in each of the different studies since there is no widely used standardized measurement procedure. For example, Chaves et al. measured the CSP at 155% of the active motor threshold and 10% of MVC grip force [26], whereas Satarnecchi considered 50% MVC as grip force and 120% of the RMT as the stimulus intensity [25]. There is evidence that measurement methods may directly affect the outcome parameters, and further to this, recent data suggest that, for example, grip force may influence the length of the CSP. However, since there is no comparative investigation, it remains speculative if these partly contradictory results are due to methodological differences, but a review of silent periods highlights the importance of standardized methods to increase reproducibility, especially in the context of the CSP [28]. Due to the high heterogeneity in the implementation and reporting of TMS studies in particular [14,27], several guideline and recommendation papers have recently been published that recommend the standardization of TMS measurements and reporting [14,35,57].

In our study, the significantly increased duration of the CSP in MS patients was constant across different stimulus intensities, which has never been demonstrated before and underscores the robustness of our findings. The increased duration of the CSP in MS, which is not due to demographic variables like age, can therefore be interpreted as a higher rate of cortical inhibitory interneurons in the patient cohort, as the CSP is mediated by gamma-aminobutyric acid (GABA), particularly by GABAB receptors within the primary motor cortex [14,28].

There is substantial evidence indicating pathological alterations in neurotransmitters like GABA and glutamate in MS [56,58,59]. It is stated that changes in glutamate metabolism could potentially play a role in the pathophysiology of MS by leading to excitotoxic damage. The GABAergic system plays a pivotal role in regulating neural network activity and plasticity [56]. A dysregulated interaction between glutamate and GABA in MS leads to synaptic loss accountable for a multitude of clinical impairments [56]. Hence, neurophysiological techniques such as TMS, with their ability to noninvasively assess alterations in these neurotransmitters, are gaining recognition for their diagnostic and therapeutic potential [14,60]. Consequently, there is a prevalent hypothesis that modulating GABAergic and glutamatergic transmission holds promise as a strategy for addressing fatigue and cognitive impairment in MS [16,18], and neurophysiological methods such as repetitive TMS (rTMS) or transcranial direct current stimulation (tDCS) are increasingly being used for the treatment of neuropsychological symptoms like fatigue in MS [18] as well as neurological and psychiatric disorders [60,61,62]. Our results are in line with the previous works and implicate that GABAergic modulation might be a therapeutic target for the modulation of fatigue in MS.

Following the abovementioned aspects of methodological heterogeneity, it is not surprising that different studies revealed different results. An association between the CSP and fatigue could be observed in at least a few studies [22,25,26], with either a decreased [25] or increased [25,26] CSP duration in MS patients and also a modulating effect of medication [25] and exhaustion [22]. The aim of our study was to investigate the association between the FSMC, a widely used self-report score to quantify fatigue [8], and the duration of the CSP. Contrary to our hypothesis, we were not able to demonstrate this association with the FSMC total score or the motor subscore. This missing association despite the difference between patients and controls might have several reasons. First, we did not measure fatigue during a particular fatiguing task but rather the subjective rating of the perceived fatigue, which might not have the same underlying causes. We chose the FSMC as it is a widely used questionnaire with high sensitivity and specificity in detecting fatigue in MS [8]. Second, it is well known that other individual demographic or disease-related variables like age, sex, or depression [10]; cortical and subcortical white and grey matter alterations [54]; and environmental conditions like the outdoor temperature [44] also affect the subjective rating of fatigue, which might also explain the missing association. Third, we cannot explicitly rule out the effect of the immunomodulatory drugs on the CSP. There is limited evidence that, for example, fingolimod affects TMS parameters [63]. No data exist on whether any immunomodulatory drug interacts with the CSP, as it also has been shown for amantadine, for example [25].

Our study has several limitations. First, we investigated the neurophysiological causes of fatigue but are aware of the differences between fatigue and fatigability [3,10]. Most of the existing studies assessed fatigability during a special task, whereas we associated the subjective rating of fatigue and the CSP, which has never been considered before. Future studies have to focus on fatigability but with recommended neurophysiological methods to guarantee validity and reproducibility. Second, our cohort includes a small sample of patients. With an exploratory approach and differences in methodology, a power calculation for a minimum sample size was not possible, but comparable TMS studies also had similar sample sizes [18,64]. Third, the aim of our study was to investigate the relationship between fatigue and the CSP in MS patients, which is why we only examined the FSMC in the patients but not the controls. Thus, we cannot explicitly exclude fatigue in the control group, but this does not affect the lack of a significant association in the patient group.

In our study, a significant difference in the duration of the cortical silent period (CSP) was found between patients with multiple sclerosis (MS) and healthy controls. This difference persisted at different stimulus intensities, extending the findings from previous work. This prolonged duration is attributed to increased GABAergic inhibitory levels in MS patients. Surprisingly, this prolonged CSP duration did not correlate with the subjective degree of fatigue, which was determined using the Fatigue Scale for Motor and Cognitive Functions (FSMC). These findings suggest a complex relationship between neurophysiological factors, such as GABAergic inhibition, and the subjective perception of fatigue in MS patients, thus challenging the conventional understanding of the mechanisms underlying MS-related fatigue.

## 5. Conclusions

In this study, we confirmed the MS-related difference in GABAergic CSP but were not able to associate the CSP with fatigue. Nevertheless, we believe that the CSP can help in the assessment of MS-associated clinical impairments, as TMS offers a unique window into the brain’s functioning as a measurement of cortical excitability. TMS measurements can add objective information even in subjective variables like fatigue and therefore contribute to diagnostics even at the individual patient level. Existing methodological recommendations must be followed and adhered to in order to validate current studies and generate a sufficient amount of comparable evidence to achieve that goal. Additionally, there is a need for a more comprehensive dataset gathered using standardized measurement protocols to explore the connection between TMS and clinical impairments in both healthy and disease states [65].

## Figures and Tables

**Figure 1 brainsci-14-00028-f001:**
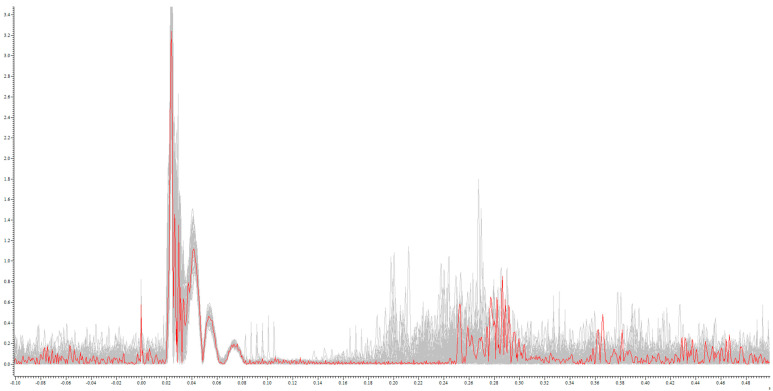
Example rectified CSP from an MS patient: single CSP during 200% RMT stimulation (red) on the sum of the CSP of all stimulations (grey): *x*-axis in mV; *y*-axis in s.

**Figure 2 brainsci-14-00028-f002:**
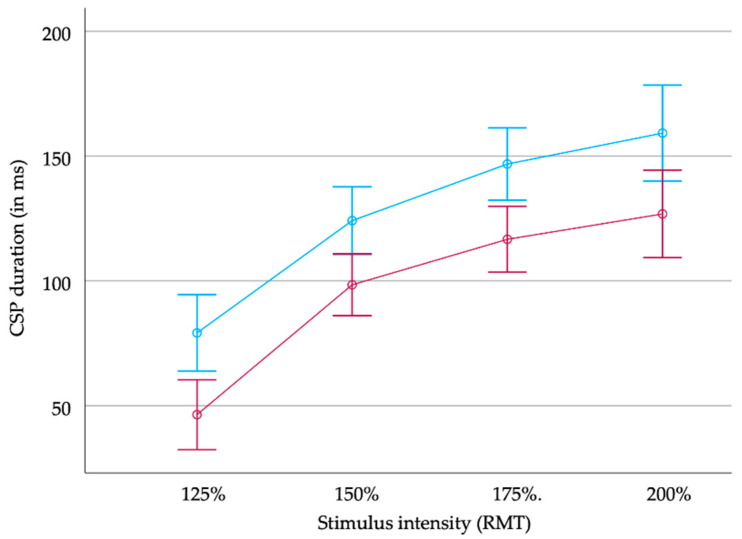
CSP duration (in milliseconds (ms)) in MS patients (upper blue line) and healthy controls (red line) in relation to stimulus intensity (resting motor threshold, RMT). Error bars indicate +/− two standard errors.

**Table 1 brainsci-14-00028-t001:** Overview of the study population. ^1^ Expanded Disability Status Scale (EDSS); ^2^ Beck’s Depression Inventory (BDI); ^3^ Fatigue Scale for Motor and Cognitive Functions (FSMC); ^4^ resting motor threshold (RMT) in % maximal stimulator output; ^5^ maximal voluntary contraction (MVC).

	Patients	Healthy Controls
Age yrs (mean (SD))	44.0 (10.7)	42.4 (11.8)
Gender male/female	9 M/14 F	9 M/14 F
Handedness right/left	21 R/2 L	21 R/2 L
EDSS (median (range)) ^1^	1.5 (0–6)	-
BDI (mean (SD)) ^2^	9.2 (6.9)	-
FSMC total (mean (SD)) ^3^	52.3 (20.8)	-
FSMC motor (mean (SD))	25.1 (10.8)	-
FSMC cognitive (mean (SD))	27.1 (10.8)	-
RMT (mean (SD)) ^4^	48.9 (7.3)	49.5 (7.7)
MVC (mean (SD)) ^5^	37.6 (15.3)	39.5 (17.4)

**Table 2 brainsci-14-00028-t002:** Matrix of Pearson correlations between CSP duration and Fatigue Scale for Motor and Cognitive Functions (FSMC) for different stimulus intensities (% resting motor threshold (RMT)), the mean of all measurements, and the ratio of the amplitude of motor-evoked potential (cMEP) and the CSP duration. No correlation was significant.

Correlation Matrix		FSMC	
CSP duration	total	motor	cognitive
125% RMT	0.144 (*p* = 0.56)	0.226 (*p* = 0.35)	0.045 (*p* = 0.85)
150% RMT	0.109 (*p* = 0.65)	0.131 (*p* = 0.58)	0.080 (*p* = 0.74)
175% RMT	−0.009 (*p* = 0.97)	0.031 (*p* = 0.89)	−0.049 (*p* = 0.83)
200% RMT	−0.236 (*p* = 0.39)	−0.300 (*p* = 0.28)	−0.158 (*p* = 0.56)
Overall RMT	−0.042 (*p* = 0.86)	−0.006 (*p* = 0.98)	−0.074 (*p* =0.75)
Ratio cMEP_amp_/CSP_duration_	0.170 (*p* = 0.50)	−0.284 (*p* = 0.25)	−0.027 (*p* = 0.92)

## Data Availability

Data are available upon request from the corresponding author (M.G.).

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
