# Peer review of "The Cortical Silent Period and Its Association with Fatigue in Multiple Sclerosis: The Need for Standardized Data Collection"

_brainsci, 2023, doi:10.3390/brainsci14010028_

Round 1

Reviewer 1 Report

Comments and Suggestions for Authors

In the current manuscript, Strauss et al., showed CSP was significantly increased in MS patients when compare to healthy controls. The results were clean and solid. I only have one comment: can the authors obtain more patient data to show the relationship between the duration of CSP and FSMC levels? 23 patients may not sufficient to establish that correlation. 

Author Response

In the current manuscript, Strauss et al., showed CSP was significantly increased in MS patients when compare to healthy controls. The results were clean and solid. I only have one comment: can the authors obtain more patient data to show the relationship between the duration of CSP and FSMC levels? 23 patients may not sufficient to establish that correlation.

Response:  Thank you for your evaluation. We fully acknowledge the relatively small sample size in our study, as explicitly addressed in the limitation section on page 8. Given the exploratory nature of our investigation, conducting a power calculation for determining a minimum sample size in advance was not feasible. It is noteworthy that comparable transcranial magnetic stimulation studies have also employed similar sample sizes. Nevertheless, we concur with the conclusion that there is a requirement for a more extensive dataset obtained through standardized measurement protocols in a larger cohort, as emphasized on page 8. For a better characterization of the cohort, we also added some more information about BDI on page 3, as well as in the results of an additional association between CSP and BDI on page 6.

Reviewer 2 Report

Comments and Suggestions for Authors

Thank you for submitting your manuscript. 

Having carefully reviewed your work, I would like to provide you with some feedback and suggestions for improvement. 

1) You don't need to write the full wording of Cortical Silent Period (CSP) after abbreviating it for the first time in the introduction line 53.

2) Materials and Methods line 107, you wrote: "Twenty subjects were right-handed," but in Table 1, you mentioned 21 R!! Decide which one is correct and modify the other.

3) The assessment of depression was not mentioned anywhere in the abstract or the introduction. The first time, depression was mentioned in the Materials and Methods without justifying its use! Make sure that you justify using the BDI from the introduction. 

4) The authors could improve the quality of the study by justifying their use of TMS parameters and compare them to previously used ones in previous studies in the discussion section.

Author Response

Having carefully reviewed your work, I would like to provide you with some feedback and suggestions for improvement. 

 Response: Thank you for your feedback with improved our manuscript.

You don't need to write the full wording of Cortical Silent Period (CSP) after abbreviating it for the first time in the introduction line 53.

Response: We have now replaced the terminology "Cortical Silent Period" throughout the manuscript.

Materials and Methods line 107, you wrote: "Twenty subjects were right-handed," but in Table 1, you mentioned 21 R!! Decide which one is correct and modify the other.

Response: Thank you, we corrected the Methods: “Twenty-one subjects were right-handed,”

The assessment of depression was not mentioned anywhere in the abstract or the introduction. The first time, depression was mentioned in the Materials and Methods without justifying its use! Make sure that you justify using the BDI from the introduction. 

Response: Thank you for that comment. We stated the importance of depression in the context of fatigue and insert new citations

  • Sparasci D, Gobbi C, Castelnovo A, Riccitelli GC, Disanto G, Zecca C, Manconi M. Fatigue, sleepiness and depression in multiple sclerosis: defining the overlaps for a better phenotyping. J Neurol 2022;269(9):4961-4971
  • Carta MG, Paribello P, Anastasia A, De Berardis D, Nardi AE, Fornaro M. Pharmacological management of depression in patients with multiple sclerosis. Expert Opin Pharmacother. 2018 Oct;19(14):1533-1540. doi: 10.1080/14656566.2018.1516207. Epub 2018 Sep 12. PMID: 30207800.)

We now wrote “ Despite its prevalence and impact, measuring fatigue objectively has remained a challenge in the field of MS research especially since there is a strong overlap to depression and sleepiness as well as associations to  other psychiatric disorder.”

We also added: “Mean BDI value was 9.24 (range: 2-28). Only two patients showed  moderate clinical relevant depressive symptomes, as indicated by the Beck's Depression Inventory (score > 19)”  to the methods section (page3)

 and “As expected fatigue measurements and BDI were highly associated (FSMC total: r = 0.721, FSMC motor subscore: r = 0.664 and FSMC cognition subscore r = 0.727 all p = <0.001.” in the result section on page 5).

We further stated that there was no correlation of electrophysiological measurements and depression score on page 5.

We added following statement to the discussion section (page 7): “Recent studies demonstrated not only a strong overlap of fatigue and depression  but also a mutually reinforcing relationship between fatigue and depression contributing to a diminished quality of life.” and implemented a new citation:

Rodgers S, Manjaly ZM, Calabrese P, Steinemann N, Kaufmann M, Salmen A, Chan A, Kesselring J, Kamm CP, Kuhle J, Zecca C, Gobbi C, von Wyl V, Ajdacic-Gross V. The Effect of Depression on Health-Related Quality of Life Is Mediated by Fatigue in Persons with Multiple Sclerosis. Brain Sci 2021;11(6).

Reviewer 3 Report

Comments and Suggestions for Authors

Abstract is written a bit more like Introduction. Although, there are sentences that belong to Abstract, this section should be rewritten.

The title does not reflect the main part of the paper, only Discussion. Hence, the paper should be rewritten or the title changed. 

Which software did you use for ANOVA?

Did you check math conditions for using ANOVA? ANOVA is not applicable in all cases.

You reference section should be improved with more recent references from 2022-23. There are references to assessment of MEP in MS also in other journals. 

You should state exactly what are the scientific contributions of your paper. Is there a contribution in argumentation of "need for standardization" or in experimental results or both? What is new for science?

Your hard job in experiments has no significance for the title's topic. Arguments for need of standardization could be found in references. It seems that you could write 2 papers - one need for standardization, and another for report of experiments and data analysis. However, these two topics should be better integrated in the single manuscript.

Data Collection is mentioned only in the title. You should have at least one section about that.

Author Response

Abstract is written a bit more like Introduction. Although, there are sentences that belong to Abstract, this section should be rewritten.

Response: We have now rewritten the entire abstract.

The title does not reflect the main part of the paper, only Discussion. Hence, the paper should be rewritten or the title changed. 

Response: We have now modified the title to emphasize the underlying rationale of our manuscript. New title: Cortical silent period and its association to fatigue in Multiple Sclerosis: the Need for Standardized Data Collection”

Which software did you use for ANOVA?

Statistical analysis was performed in SPSS Statistics 22 (IBM, Armonk, NY, USA, https://www.ibm.com/analytics/spss-statistics-software) we no stated this in the method section (page 5)

Did you check math conditions for using ANOVA? ANOVA is not applicable in all cases.

We checked the preequisites for an ANOVA visually by histograms and stated the following in the method section. „Since electrophysiological data met prerequisites for an ANOVA we used a repeated-measures (rm) ANOVA model with ‘stimulus ‘intensity’ (4 intensities) as within factor and group as between group factor (healthy vs. patients) to test the first hypothesis.” (page 4).

You reference section should be improved with more recent references from 2022-23. There are references to assessment of MEP in MS also in other journals. 

You should state exactly what are the scientific contributions of your paper. Is there a contribution in argumentation of "need for standardization" or in experimental results or both? What is new for science?

Response:

We thank the author for that comment. We added very recent references from 2023.

Leodori, M. Mancuso, D. Maccarrone, M. Tartaglia, A. Ianniello, F. Certo, et al.Neurobiol Dis 2023 Vol. 180 Pages 106073. Neural bases of motor fatigue in multiple sclerosis: A multimodal approach using neuromuscular assessment and TMS-EEG.

We also have added a complete paragraph in the discussion about the need for standardization and its importance for science (page 7).

Your hard job in experiments has no significance for the title's topic. Arguments for need of standardization could be found in references. It seems that you could write 2 papers - one need for standardization, and another for report of experiments and data analysis. However, these two topics should be better integrated in the single manuscript.

Data Collection is mentioned only in the title. You should have at least one section about that.

Response: Thank you for your feedback. However, we maintain the perspective that both topics are well-suited for inclusion in the manuscript, as we present a clear scientific hypothesis addressing areas in electrophysiological research that require improvement to yield meaningful insights. Our results suggest no association with self-reported fatigue, either indicating an unexpected outcome or revealing a limitation in the generation of the hypotheses based on the existing heterogeneous literature. This underscores the necessity of standardized measurement protocols to confidently generate hypotheses and comparable results. If we were to split this into two articles, we would contribute only one additional result regarding the heterogeneous literature and another narrative description emphasizing the necessity of standardized measurements without a direct connection to a define scientific hyoptheses.

We added a complete paragraph about the need for standardization on page 7.

Reviewer 4 Report

Comments and Suggestions for Authors

In the present study the Authors aimed to evaluate two primary objectives. Firstly, the Researchers tried to confirm the disparities in Cortical Silent Period (CSP) duration between patients diagnosed with multiple sclerosis and a healthy control group. This comparative analysis served as a fundamental step towards understanding the neurological basis of fatigue in multiple sclerosis (MS). Secondly, the Researcher tried to demonstrate a clear relationship between the duration of CSP and Fatigue Scale for Motor and Cognitive Function (FSMC) levels in people with MS. By investigating this relationship, the Researchers aimed to provide insight into the complex interplay between cortical interhemispheric excitability and the subjective experience of fatigue as described by the FSMC. This multi-faceted approach not only might deepen the understanding of the neural mechanisms involved in MS-related fatigue, but can also provide an objective perspective on how cognitive and physical aspects of fatigue are represented at the cortical level, which could pave the way for more targeted and effective interventions in the future.

Overall, I found the study timely, original, well conducted and scientifically sound. However, I have some comments aimed at improving the quality of the paper, and these are outlined below:

1. In the introduction, a brief note on the fact that MS often presents a comorbidity with psychiatric disorders (please see and refer to following doi: 10.1080/14656566.2018.1516207).

2. How many subjects that were screened to reach the final sample of 23 RRMS patients, were not included for any reason (why?) or refused to participate? Please, add some more information on this point.

3. Were the presence of psychiatric disorder and/or drug abuse (the latter history or current presence) evaluated, and how?

4. As well was the presence of an intellectual disability assessed and how?

5. The severity of depression was evaluated using the BDI, but the term "depression" is inappropriate as BDI evaluates "depressive symptoms". The clinical diagnosis of depression requires also an accurate direct assessment.

6. Table 1 can be omitted and briefly condensed in the text.

7. Translating into “real world” clinical practice and medicine, what possible clinical shreds of evidence might arise from the present study and what the Researchers do suggest improving clinical practice? Please add a brief paragraph on possible suggestions in terms of integrative care. I suggest expanding a bit and make deeper and clear the "conclusions" section.

Author Response

Overall, I found the study timely, original, well conducted and scientifically sound. However, I have some comments aimed at improving the quality of the paper, and these are outlined below:

Response: Thank you for enhancing the quality of our manuscript with your input.

In the introduction, a brief note on the fact that MS often presents a comorbidity with psychiatric disorders (please see and refer to following doi: 10.1080/14656566.2018.1516207).

 Response: Thank you we stated the importance of depression in the context of fatigue and insert new citations

  • Sparasci D, Gobbi C, Castelnovo A, Riccitelli GC, Disanto G, Zecca C, Manconi M. Fatigue, sleepiness and depression in multiple sclerosis: defining the overlaps for a better phenotyping. J Neurol 2022;269(9):4961-4971
  • Carta MG, Paribello P, Anastasia A, De Berardis D, Nardi AE, Fornaro M. Pharmacological management of depression in patients with multiple sclerosis. Expert Opin Pharmacother. 2018 Oct;19(14):1533-1540. doi: 10.1080/14656566.2018.1516207. Epub 2018 Sep 12. PMID: 30207800.)

We now wrote “ Despite its prevalence and impact, measuring fatigue objectively has remained a challenge in the field of MS research especially since there is a strong overlap to depression and sleepiness as well as associations to  other psychiatrc disorders.”

How many subjects that were screened to reach the final sample of 23 RRMS patients, were not included for any reason (why?) or refused to participate? Please, add some more information on this point. 

Were the presence of psychiatric disorder and/or drug abuse (the latter history or current presence) evaluated, and how?

As well was the presence of an intellectual disability assessed and how? 

 Response: We did not document systematically how many patients were asked in the outpatient clinic that were not able to take part due to exclusion criteria or refused to participate. We now added more information in the method section about exclusion criteria (page 3): “Patients with other neurological or psychiatric diseases as well as potential interacting comedication (i.e. antispastic or anticonvulsive medication) were excluded.“ 

The severity of depression was evaluated using the BDI, but the term "depression" is inappropriate as BDI evaluates "depressive symptoms". The clinical diagnosis of depression requires also an accurate direct assessment.

Response: thank you, we have revised the wording in accordance with your recommendations.

Table 1 can be omitted and briefly condensed in the text. 

Resonse: We find tables to be somewhat more concise for summarizing data at a glance.

Translating into “real world” clinical practice and medicine, what possible clinical shreds of evidence might arise from the present study and what the Researchers do suggest improving clinical practice? Please add a brief paragraph on possible suggestions in terms of integrative care. I suggest expanding a bit and make deeper and clear the "conclusions" section

Resonse: Thank you for that comment. We now rewrote the entire conclusion (page 9). 

Round 2

Reviewer 3 Report

Comments and Suggestions for Authors

1) AUTH.ANS: "We checked the preequisites for an ANOVA visually by histograms and stated the following in the method section."

 You should add data in the manuscript. Are your data obey "normal law" (Gaussian distribution)?

2.) AUTH. ANS: "However, we maintain the perspective that both topics are well-suited for inclusion in the manuscript, as..."

I don't say that you cannot have both in one manuscript. However, it should be better presented that reader could easily follow your flow.

Author Response

Point by point reply:   reviewer: You should add data in the manuscript. Are your data obey "normal law" (Gaussian distribution)?   response: We added some more information on page 5: "Data were normally distributed as determined with visual inspection as well as Kolmogorov- Smirnov tests, so we used parametric statistical procedures. A repeated-measures (rm) ANOVA model with ‘stimulus ‘intensity’ (4 intensities) as within factor and group as between group factor (healthy vs. patients) was used to test the first hypothesis."   Reviewer: I don't say that you cannot have both in one manuscript. However, it should be better presented that reader could easily follow your flow.   Response: We added more information in the introduction (page 2), as the reviewer stated. We now hope that both topics are introduced sufficiently: "In the heterogeneous literature on the association between neurophysiology and fatigue, it should also be noted that the TMS methods used are differing even for the same outcome variable. CSP studies differ to varying degrees, for example in the stimulus intensity, which reduces the comparability of the results. This methodological problem is well known and has recently been addressed by a number of guidelines."